# Composite Polybenzimidazole Membrane with High Capacity Retention for Vanadium Redox Flow Batteries

**DOI:** 10.3390/molecules26061679

**Published:** 2021-03-17

**Authors:** Jacobus C. Duburg, Kobra Azizi, Søren Primdahl, Hans Aage Hjuler, Elena Zanzola, Thomas J. Schmidt, Lorenz Gubler

**Affiliations:** 1Electrochemistry Laboratory, Paul Scherrer Institut, CH-5232 Villigen, Switzerland; jacobus.duburg@psi.ch (J.C.D.); thomasjustus.schmidt@psi.ch (T.J.S.); lorenz.gubler@psi.ch (L.G.); 2Blue World Technologies, Egeskovvej 6C, DK-3490 Kvistgård, Denmark; kaz@blue.world (K.A.); spr@blue.world (S.P.); hah@blue.world (H.A.H.); 3Danish Center for Energy Storage, Frederiksholms Kanal 30, DK-1220 Copenhagen K, Denmark; 4Laboratory for Physical Chemistry, ETH Zurich, CH-8093 Zurich, Switzerland

**Keywords:** polybenzimidazole, polypropylene, skin layer, support layer, interlocking interface, composite asymmetric membrane, vanadium redox flow battery, discharge capacity, capacity retention

## Abstract

Currently, energy storage technologies are becoming essential in the transition of replacing fossil fuels with more renewable electricity production means. Among storage technologies, redox flow batteries (RFBs) can represent a valid option due to their unique characteristic of decoupling energy storage from power output. To push RFBs further into the market, it is essential to include low-cost materials such as new generation membranes with low ohmic resistance, high transport selectivity, and long durability. This work proposes a composite membrane for vanadium RFBs and a method of preparation. The membrane was prepared starting from two polymers, *meta*-polybenzimidazole (6 μm) and porous polypropylene (30 μm), through a gluing approach by hot-pressing. In a vanadium RFB, the composite membrane exhibited a high energy efficiency (~84%) and discharge capacity (~90%) with a 99% capacity retention over 90 cycles at 120 mA·cm^−2^, exceeding commercial Nafion^®^ NR212 (~82% efficiency, capacity drop from 90% to 40%) and Fumasep^®^ FAP-450 (~76% efficiency, capacity drop from 80 to 65%).

## 1. Introduction

Among electrochemical energy storage technologies, redox flow batteries (RFBs) are very flexible. In particular, they have the unique advantage of decoupling energy storage from power output. Energy is stored in two external electrolyte-containing tanks, and charge/discharge reactions take place in an electrochemical reactor where the electroactive species, dissolved in the electrolyte solutions, undergo reversible redox reactions on the surface of porous carbon-based electrodes. Inside the electrochemical cell, a membrane is used as a separator between the two porous electrodes, functioning as a polymer electrolyte for selective transport of dissolved ionic species. 

Specifically, membranes in a unit electrochemical cell, incorporated in an RFB setup, have the primary role of mediating the transport of ionic charges between the negative and the positive cell compartment, to allow each half-cell redox reaction to proceed in a continuous manner while electrons are exchanged via an external circuit [1]. At the same time, it is not desirable for the electroactive species to transfer across the membrane, as this may impair the coulombic efficiency, depolarize an electrode, pollute the product, reduce capacity retention, or cause other unwanted effects. 

A large number of publications have focused on membranes for RFBs [2,3,4], where researchers proposed new designs and materials to increase stability and efficiency in this storage technology during prolonged charge/discharge cycling. Recent studies showed membranes consisting of an asymmetric architecture [5,6,7] and others comprising a porous support coated with a thin polymer layer to act as a barrier against the crossover of electroactive species (e.g., vanadium or organic molecules) [8,9,10]. Furthermore, the membrane design can become relevant on a technical scale in terms of investment costs. Indeed, if the membrane has a low ohmic resistance, the cell current density can be enhanced at a given round trip efficiency, leading to an increased power density. Accordingly, this can play a role in the size of the RFB stack, potentially leading to a reduction in the investment costs [6]. One method of reducing the ohmic resistance of the membrane is by reducing its thickness [5,11,12]. However, polymer electrolytes with a thickness in the micron range do not exhibit the mechanical strength to be used as free-standing films due to the risk of rupturing or developing pinholes during handling, cell assembly, and operation. To overcome this problem, the thin polymer electrolyte film can be strengthened with a porous support to provide mechanical robustness while allowing the access of a liquid electrolyte through its porous structure.

In terms of materials, membranes prepared from a polybenzimidazole (PBI) type polymer are receiving attention in the scientific community for their diverse applicability in various fields and uses, such as high-temperature proton exchange membrane fuel cells, electrodialysis, and organic solvent nanofiltration [13,14,15,16,17], including RFBs and, in particular, vanadium RFBs (VRFBs) [18,19,20,21]. Whereas commercial Nafion^®^ is a cation exchange membrane (CEM), PBI in its pristine state is an insulator, but it develops the properties of an anion exchange membrane (AEM) in acidic conditions due to the uptake of electrolyte and the subsequent protonation of the nitrogen atoms in the imidazole group, thereby enhancing its barrier properties versus cations [22]. In addition, PBI’s highly chemically stable backbone is particularly suitable for the oxidative and acidic environment of the VRFB [23], and studies have confirmed that a PBI pore size range from 0.5 to 2 nm can help to reduce vanadium ion crossover [23,24,25].

In this work, a composite asymmetric membrane for VRFBs and a method of preparation are described. The membrane was prepared starting from two polymeric layers, a thin *meta*-polybenzimidazole film (*m*-PBI, 6 μm thickness), herein called the “skin layer”, and a hydrophobic mesoporous polypropylene layer TreoPore^®^ PDA-30 (PP, 30 μm thickness, non-woven, porosity >60%, Treofan, Raunheim, Germany), herein called the “support layer”, through a gluing procedure by hot-pressing. This composite membrane was tested in a VRFB cell to assess performance and cycling stability over 90 consecutive charge/discharge cycles at 120 mA·cm^−2^. The cation exchange membrane Nafion^®^ NR212 and the anion exchange membrane Fumasep^®^ FAP-450 were used as commercial benchmarks.

## 2. Results

The *m*-PBI was synthesized (see Section 4) and characterized using various techniques to study its purity and chemical stability. Proton nuclear magnetic resonance (^1^H-NMR) in dimethyl sulfoxide (DMSO)-d_6_, Fourier-transform infrared spectroscopy (FTIR), thermogravimetric analysis, and differential thermal analysis (TGA/DTA) were performed (see procedure in Section 4 and results in the Appendix A). In addition, the inherent viscosity and molecular weight of the synthesized *m*-PBI were measured (see Appendix A) according to previous work [26]. The inherent viscosity was 1.14 dL·g^−1^, and the molecular weight, 58,000 Da, was determined using the empirical Mark–Houwink constants (*K* = 1.94 × 10^−4^ dL·g^−1^ and *α* = 0.791) [27]. Later, it was demonstrated that a film of *m*-PBI remained stable over 4 months by performing an ex situ stability test in 1.6 M V(V) in 2 M H_2_SO_4_ and 0.05 M H_3_PO_4_ solution at room temperature. In this period of time, no V(IV) formed through the reaction of *m*-PBI with V(V), as confirmed by ultraviolet (UV)–visible light spectroscopy. In addition, for a film of *m*-PBI with a 39 µm thickness, the water and electrolyte uptake, the in-plane conductivity, and an accelerated stress test in a solution of 0.2 M cerium(IV) sulfate were determined and compared to those of standard commercial materials for VRFBs (Nafion^®^ NR212 and Fumasep^®^ FAP-450, respectively) (see results in the Appendix A).

Scanning electron microscopy (SEM) on a 6 µm *m*-PBI film was used as a qualitative study to investigate the morphology of the pristine polymer (Figure 1A)).

Subsequently, the polypropylene–polybenzimidazole (abbreviated as PP-PBI) composite asymmetric membrane was prepared through a gluing method by hot-pressing from the thin *m*-PBI (6 μm) film and the porous polypropylene (TreoPore^®^ PDA-30, thickness 30 μm, nonwoven, porosity > 60%, Treofan, Raunheim, Germany) layer. A schematic of the membrane preparation is displayed in Figure 2, alongside a photo of the composite membrane after the hot-pressing treatment. The detailed procedure can be found in Section 4.

The SEM image shows the thin *m*-PBI “skin layer” bonded to the porous PP “support layer” (Figure 1B). In particular, the thickness of the *m*-PBI layer looks smaller than before the gluing procedure (Figure 1A), due to the solubilization of the latter into the porous PP layer. The area in between the two layers is defined herein as the “interlocking” interface (i.e., interphase). In addition, the energy dispersed X-ray (EDX) analysis qualitatively confirmed C and N in the PP-PBI composite membrane as the most significant elements (Figure 1C,D). C was evenly distributed across the sample, while N was mostly present in and close to the *m*-PBI layer of the membrane configuration.

Later, the diffusion of V(IV) through the composite PP-PBI membrane was measured and compared to that in the commercial NR212 and FAP-450 (Figure 3). A description of the full procedure is reported in Section 4. Both the commercial membranes showed a significant V(IV) diffusion, while the PP-PBI membrane exhibited an almost negligible one.

The experimentally measured values of V(IV) diffusion are reported in Table 1 for all the three membranes.

Subsequently, the PP-PBI composite membrane was assembled and tested in a VRFB single cell of laboratory scale (25 cm^2^) (the cell components and the operation procedure, as well as the electrolyte composition, are detailed in Section 4). The cycling performance of NR212 and FAP-450 was evaluated as a comparison to the membrane of the present work. First, the in situ through-plane ohmic area resistance of the cell with the membrane at −50% electrolyte state of charge (SOC) (average oxidation state of the vanadium electrolyte equal to 3.5) was measured between 100 kHz and 0.1 Hz at zero direct current (DC) and a perturbation amplitude of 100 mV by electrochemical impedance spectroscopy (EIS) (Table 2). In the same manner, the cell resistance with only a film of hydrophobic mesoporous PP TreoPore^®^ PDA-30 (thickness 30 μm, non-woven, porosity > 60%, Treofan, Raunheim, Germany) was measured and found equal to 0.35 Ω·cm^2^. Then, the electrolyte was galvanostatically charged to 100% SOC by applying 40 mA·cm^−2^. During the initial charge, polarization curves were recorded at 20%, 30%, 50%, 70%, and 90% SOC by consecutively charging and discharging the battery for 20 s at 20, 40, 60, 80, 100, 120, and 200 mA·cm^−2^ (Figure 4). Lastly, upon fully charging the battery, five consecutive charge/discharge cycles were performed from 0.80 V to 1.65 V as the lower and upper potential limits at a current density of 40, 80, 120, 160, and 200 mA·cm^−2^ (Figure 5). The first cycle at each current density was not included in the analysis as a deviation related to the change in current density could be seen in the data.

The ohmic area resistance of the cell with PP-PBI (0.69 Ω·cm^2^) was higher than that with commercial NR212 (0.49 Ω·cm^2^), but lower than that with FAP-450 (1.28 Ω·cm^2^). The influence of the ohmic resistance was observed in both the polarization curves (Figure 4) and the cycling efficiencies (Figure 5). An increase in the resistance increased the overpotential, thus leading to a steeper polarization curve and to a lower voltaic efficiency and vice versa.

Lastly, the three membranes were galvanostatically cycled at 120 mA·cm^−2^ for 90 consecutive cycles, from 0.80 V to 1.65 V as the lower and upper potential limits, in 1.6 M V in 2 M H_2_SO_4_ and 0.05 M H_3_PO_4_ electrolyte (Figure 6). Efficiencies and discharge capacity were calculated according to Equations (5)–(8) in Section 4. The PP-PBI membrane showed a good performance (coulombic efficiency ~99%, voltaic efficiency ~84%, and energy efficiency ~84%). Importantly, this membrane exhibited a high and stable discharge capacity (~90%) over time with a 99% capacity retention, exceeding those of the commercial NR212 (capacity drop from 90% to 40%) and FAP-450 (capacity drop from 80% to 65%).

Importantly, the PP-PBI membrane after the extended galvanostatic cycling in the VRFB revealed no ruptures or delamination of the layers (Figure 7). The SEM image shows the intact *m*-PBI “skin layer” bonded to the PP “support layer” as before the cell testing (see Figure 1B).

## 3. Discussion

The asymmetric membrane of this work was made by gluing a thin *m*-PBI (6 µm) film to a porous PP (30 µm) layer by hot-pressing (see Figure 2). Whereas conventional blade casting of an *m*-PBI solution on a PP support might result in significant seepage of the *m*-PBI polymer into the porous network, resulting in blocked pores and a higher ohmic resistance, the gluing by hot-pressing method minimizes this risk by only partially solubilizing the *m*-PBI skin layer. This partial solubilization of *m*-PBI during the gluing and subsequent hot-pressing to the PP layer results in a particular region of the membrane, herein called the “interlocking” interface (i.e., interphase). Qualitative evidence of this interface was found by SEM, where a slightly thinner *m*-PBI layer (5 µm) was interconnected to the porous PP with no visible delamination (Figure 1B). The shrinking of the “skin layer” can be explained by the small quantities of PBI polymer flowing into the support layer (PP), creating the mechanically bonded interface but simultaneously thinning the “skin layer”.

The reported data represent the performance of an independently prepared membrane, with each test being conducted a minimum of two times to verify its results.

V(IV) diffusion through the composite PP-PBI membrane was found to be the lowest ((14 ± 1) × 10^−8^ cm^2^·min^−1^), while commercial Nafion^®^ NR212 suffered the highest V(IV) diffusion ((744 ± 9) × 10^−8^ cm^2^·min^−1^), likely due to large negatively charged ionic channels with a diameter of 4–5 nm [28,29]. This is in reasonable agreement with the findings in Oldenburg et al. [30]. Commercial NR212 and Fumasep^®^ FAP-450 showed a steep linear relationship between the concentration of the permeated V(IV) and the time, while the PP-PBI membrane displayed an almost linear trend (Figure 3). The trend of V(IV) diffusion through the PP-PBI membrane was quite peculiar; for the first 9 days of the experiment, no V(IV) diffusion was detected by UV/visible light spectroscopy, which developed to a very slow and stepwise growth until the end of the test. The V(IV) diffusion behavior through the PP-PBI composite membrane was not deeply studied, but it is in reasonable agreement with other works on *m*-PBI based membranes, where no V(IV) diffusion could be detected over at least 24 h [12,31]. Importantly, the diffusion of the other V species needs to be studied to obtain a more comprehensive understanding of the membrane transport properties.

The cell with the PP-PBI membrane showed a slightly higher ohmic area resistance than that with NR212 (0.69 vs. 0.49 Ω·cm^2^, respectively); however, this value was still significantly lower than that with FAP-450 (1.28 Ω·cm^2^). The influence of this slight increase in ohmic area resistance vs. NR212 can be seen in the polarization curves and in the voltaic efficiency of the PP-PBI membrane, as it led to an increase in the ohmic overpotential (η_ohm_), steepening the polarization curve and lowering the voltaic efficiency. A similar trend, but to a more extreme extent, could be seen for FAP-450.

The PP-PBI membrane cycled at different current densities showed relatively good efficiencies (from >90% at the lowest current density of 40 mA·cm^−2^ to ~75% at the highest current density of 200 mA·cm^−2^) (Figure 5A–C). Then, when charged/discharged at 120 mA·cm^−2^ for 90 consecutive cycles, it exceeded the performance of the commercial NR212 and FAP-450. In particular, it showed the greatest discharge capacity (~90%) with a 99% capacity retention. The investigation of such a stable performance in a VRFB from the PP-PBI membrane is currently in progress; however, at this stage, it is believed that the asymmetric configuration with a relatively thin *m*-PBI film supported on a porous PP layer could be a benefit. Another aspect worth mentioning related to this prolonged cycling test in the VRFB involved the thickness of the *m*-PBI film and the applied current density. In a recent study, Oldenburg et al. [22] demonstrated how the net V flux in an amphoteric PBI/Nafion bilayer membrane for VRFBs shifted to the positive side of the system by increasing the current density. Because of this shift, a thicker PBI layer (>4 μm) was needed to balance the net V flux. This study is in good agreement with the present work, where the thickness of the *m*-PBI film (6 μm) and the applied current density yielded a very low net V flux.

Lastly, the SEM characterization post cycling test qualitatively showed no delamination of the layers or ruptures in the membrane structure. Interestingly, the SEM characterization revealed a thicker *m*-PBI film (11 μm) than before the cycling in the VRFB (5 μm). This phenomenon can be attributed to the swelling of the composite membrane in the electrolyte during the extended cycling. In fact, EDX analysis confirmed the presence of S coming from the H_2_SO_4_ contained in the electrolyte, likely retained in the membrane structure despite several washing steps in Milli-Q water.

Despite the excellent performance in an electrochemical cell, low cost with respect to membranes of the Nafion^®^ type, and good adhesion between the “skin” and “support” layer of the prepared PP-PBI membrane, it has to be mentioned that membranes prepared from *m*-PBI are required to be thin, due to their lower conductivity, to compete with the performance of Nafion^®^ type membranes. As such, further research on improving the conductivity of *m*-PBI, for example, through functionalization of the imidazole group, is desired to further enhance the performance of the membrane and, thus, improve its applicability.

## 4. Materials and Methods

### 4.1. Preparation of the m-PBI

*m*-Polybenzimidazole (abbreviated as *m*-PBI) was synthesized by Blue World Technologies (BWT) from 3,3′-diaminobenzidine and isophthalic acid at a 1:1 molar ratio by polymerization in polyphosphoric acid (PPA) at 180–250 °C under N_2_ for 2 h according to a procedure by Li et al. [32]. For more details on the so-called PPA process, the reader is referred to [33,34]. This specific temperature range refers to the initial synthesis temperature (180 °C), corresponding to the addition of the PPA, with 250 °C as the used and maximum synthesis temperature to avoid ending up with a higher molecular weight. The prepared polymer solution was poured into a water bath to obtain PBI fibers. It was neutralized by addition of a NaOH solution, washed with distilled water, and dried in an oven at 120 °C.

The obtained PBI powder was dissolved in *N*,*N*-dimethylacetamide (DMAc) (10% PBI in DMAc) at room temperature and then heated to 100 °C under continuous stirring for 12 h. The membrane was prepared by casting the PBI solution onto a glass plate and allowing the solvent to evaporate slowly in a temperature range from 40 to 120 °C with a step rate of 10 °C·h^−1^. The membranes were then washed with distilled water at 80 °C. Lastly, the film was dried at 130 °C for 2 h.

### 4.2. Characterization of the m-PBI

Inherent viscosity measurements were conducted using a capillary viscometer (SI Analytics Glass Kinematic Ubbelohde viscometer 501 13/Ic, Mainz, Germany) at a concentration of 0.5 g·100 mL^−1^ PBI in sulfuric acid 96% at 30 °C.

NMR spectroscopy in DMSO-d_6_ was performed using a Bruker Ascend 400 MHz NMR spectrometer (Billerica, MA, USA).

Transmittance Fourier-transform infrared spectroscopy was carried out on pristine and acid-doped *m*-PBI films with the help of a Bruker Vertex V70 spectrometer (Billerica, MA, USA). Before analyzing the *m*-PBI samples (2.0 × 3.0 cm, 6 µm), they were dried at 60 °C under vacuum for 2 h to remove residual water in the polymer. For the acid-doped *m*-PBI, before drying, the samples were placed in a 2 M H_2_SO_4_ solution for 18 h and subsequently dipped five times in deionized water to remove any H_2_SO_4_ present on the surface of the polymeric film.

To study the thermal stability of the membranes, TGA/DTA was conducted using a TGA 550 (TA instruments Inc., New Castle, DE, USA) with a heating rate of 10 °C·min^−1^ from room temperature to 950 °C.

The chemical stability of the *m*-PBI layer (ex situ) was investigated by immersing a PBI membrane piece (3 × 3 cm) for 4 months in 50 mL of vanadium solution (1.6 M V(V)) in 2.0 M H_2_SO_4_ and 0.05 M H_3_PO_4_, obtained by charging the electrolyte solution V(IV):V(III) (50:50) from Oxkem (Reading, UK) in a VRFB cell (catholyte), at room temperature.

### 4.3. Preparation of PP-PBI Composite Asymmetric Membrane

A *meta*-polybenzimidazole film (*m*-PBI, 6.5 × 6.5 cm, thickness 6 μm), prepared as described in Section 4.1, was placed on a polytetrafluoroethylene sheet (PTFE, 15.5 × 15.5 cm, 200 μm thickness, Itin Technik GmbH, Twann, Switzerland) and covered by a microporous polypropylene separator (PP, TreoPore^®^ PDA-30, 7.0 × 7.0 cm, thickness 30 μm, porosity > 60%, Treofan, Raunheim, Germany). Then, using Kimwipe^®^ disposable wipers (Kimberly-Clark Professional, Koblenz, Germany), the PP separator was wetted by a gluing solution comprising dimethylacetamide (DMAc, 99%, Alfa Aesar, Kandel, Germany) in isopropanol (IPA, 99.5%, Carl Roth, Karlsruhe, Germany) with a volume ratio of 1:2. Simultaneously, air between the layers was removed until the underlying *m*-PBI layer was seen clearly. Subsequently, the excess of solvent was removed and the stack was completed by placing two regular tissues (10.5 × 10.5 cm) and a PTFE sheet (15.5 × 15.5 cm, 200 μm thickness, Itin Technik GmbH, Twann, Switzerland) on top of the PP film. The stack was hot-pressed at 80 °C with a force of 2.5 tons for 15 min. Lastly, the hot-pressed composite membrane was placed in a vacuum oven (Gallenkamp, Loughborough, UK) at 100 °C for 45 min and then in IPA for 1 h. The resulting composite membrane was then ready for further tests or characterization measures without any additional pretreatment.

### 4.4. Scanning Electron Microscopy (SEM) and Energy Dispersed X-ray (EDX) Analysis

Scanning electron microscopy (SEM) cross-sections of thin pristine *m*-PBI (6 μm thickness) and the corresponding composite PP-PBI membrane were prepared using a liquid nitrogen breaking method. First, the specimen (0.5 × 2.0 cm) was wetted in IPA and then immersed in liquid nitrogen for approximately 20 s with the help of a pair of tweezers. Next, the sample was broken into two pieces and the cross-sections were placed face up in a slotted specimen stub (12 mm diameter, Agar Scientific, Stansted, UK). Lastly, the SEM samples were sputter-coated with a 10 nm layer of chromium using a LEICA EM ACE600 coater (Leica Microsystems, Heerbrugg, Switzerland).

SEM images of the thin *m*-PBI layer (6 μm thickness) and of the prepared composite PP-PBI membrane cross-section samples before and after cycling tests in the VRFB were obtained using a Hitachi Regulus 8230 series high-resolution scanning electron microscope (Tokyo, Japan), equipped with an energy dispersed X-ray analysis (EDX) detector. Experimental conditions of 6.0 kV accelerating voltage and 2 to 3 μA current were used for both electron imaging and EDX analysis. Secondary electron (SE) images were recorded with an in lens detector at a working distance (WD) of 8 to 9 mm depending on the sample. The built-in software “Hitachi Regulus” was used for SEM imaging and the software “Oxford-Aztec 3.3” was used for EDX analysis.

### 4.5. Vanadium RFB Relevant Membrane Properties

#### 4.5.1. Water and Electrolyte Uptake

Water and electrolyte uptake was determined for the following materials: (i) *m*-PBI (2.0 × 4.0 cm, 39 μm thickness, Blue World Technologies, Kvistgård, Denmark), (ii) Fumasep^®^ FAP-450 (2.0 × 4.0 cm, Fumatech BWT Group, Bietigheim-Bissingen, Germany), and (iii) Nafion^®^ NR212 (2.0 × 4.0 cm, Ion Power, New Castle, DE, USA).

The dry weight of the membrane (*w*_dry_) was obtained after drying it under vacuum at 55 °C for 22 h. The weight measurement was carried out in a closed vial to limit the uptake of moisture from the air. Then, the weight of the membrane in the wet state (*w*_wet_) was determined after immersion for 2 days in deionized water or in 1.6 M vanadium in 2 M H_2_SO_4_ and 0.05 M H_3_PO_4_ electrolyte (SOC −50%, 3.5 oxidation state, Oxkem, Reading, UK), followed by the removal of droplets on the surface with a tissue. In this case, the wet weight was measured in a vial to reduce the evaporation of water from the membrane. Lastly, water and electrolyte uptake of pristine *m*-PBI and of commercial membranes NR212 and FAP-450 was calculated according to Equation (1).
(1)$$\mathrm{Uptake}=\frac{w_{wet}-w_{dry}}{w_{dry}}\cdot 100\%$$

#### 4.5.2. In-Plane Conductivity

The in-plane conductivity in 1.6 M vanadium in 2 M H_2_SO_4_ and 0.05 M H_3_PO_4_ vanadium electrolyte (SOC −50%, 3.5 oxidation state, Oxkem, Reading, UK) was determined for (i) *m*-PBI (4.0 × 5.0 cm, 39 μm thickness, Blue World Technologies, Kvistgård, Denmark), (ii) Fumasep^®^ FAP-450 (4.0 × 5.0 cm, Fumatech BWT Group, Bietigheim-Bissingen, Germany), and (iii) Nafion^®^ NR212 (4.0 × 5.0 cm, Ion Power, New Castle, DE, USA). All membranes were immersed in the electrolyte for 2 days. Then, samples with a size of 1.2 × 3.0 cm were punched out. These were immersed again in the electrolyte for another 2 h. Subsequently, the membrane thickness was measured using a thickness gauge (Heidehain MT12B, Measurement Technologies & Supply Inc, St. Clair Shores, MI, USA) and the resistance was recorded in air using a four-point probe setup (Bekktech BT-112 cell, Scribner, Southern Pines, NC, USA, and LCR-6100, Gw Instek, New Taipei City, Taiwan, LCR meter). Lastly, the in-plane conductivity (σ in mS·cm^−1^) was calculated according to Equation (2), where *l* is the distance between the sense probes in mm, *R* is the measured resistance in kΩ, *w_sample_* is the width of the sample in cm, and *t_wet_* is the wet thickness in µm.
(2)$$\sigma =\frac{l}{R\cdot w_{sample}\cdot t_{wet}}$$

#### 4.5.3. V(IV) Diffusion

Diffusion of vanadium(IV) through the membrane was determined in a home-built cell (Appendix A). The flasks were filled on one side with 150 mL of 1.6 M VOSO_4_ (>95% VOSO_4_·5 H_2_O, Thermo Fisher Scientific, Reinach, Switzerland) in 2 M H_2_SO_4_ and on the other side with 150 mL of 1.6 M magnesium sulfate (MgSO_4_, anhydrous, Reagent Plus^®^, ≥99.5%, Sigma-Aldrich, Buchs, Switzerland) in 2 M H_2_SO_4_ as the enriched and deficient compartments, respectively. Each compartment was continuously stirred to avoid any deviations in concentration. This test was carried out for (i) the PP-PBI membrane of the present work (5.0 × 5.0 cm, comprising of a 6 μm *m*-PBI layer and a 30 μm PP layer), (ii) Fumasep^®^ FAP-450 (5.0 × 5.0 cm, Fumatech BWT Group, Bietigheim-Bissingen, Germany), and (iii) Nafion^®^ NR212 (5.0 × 5.0 cm, Ion Power, New Castle, DE, USA). For each membrane, the permeability test lasted a minimum of 220 h. During this period, UV/visible light measurements between 200 nm and 800 nm were performed, from which the VOSO_4_ absorbance peak at 765 nm was used to calculate the VOSO_4_ concentration. The measurements were carried out by filling two quartz cuvettes (Hellma Analytics, Zumikon, Switzerland) with 2.5 mL of solution from the MgSO_4_ flask. Each time, the measured solution was transferred back to the VOSO_4_ flask to avoid significant volume changes. In the case of NR212, after 4 days, the samples were diluted five times to ensure that an absorbance of 1.2 was not exceeded. Therefore, 0.5 mL from the receiving flask was diluted with 2 mL of 1.6 M MgSO_4_ in 2 M H_2_SO_4_ to prepare the measurement solutions, and 0.5 mL was removed from the enriched compartment to minimize hydrostatic pressure effects.

The VOSO_4_ diffusion through the membrane (abbreviated as *D_VOSO4_* in cm^2^·min^−1^ in Equation (3)) was calculated according to the Fick’s Law [31,35]. *V_D_* is the solution volume in the MgSO_4_ flask (in mL), *A* is the exposed area of the membrane (7.07 cm^2^), *L* is the thickness of the membrane in the swollen state (in cm), *C_E_* is the VOSO_4_ concentration in the enriched compartment (in M), and *C_D_* is the measured VOSO_4_ concentration in the deficient compartment (in M). *C_D_* at a given time was obtained by a linear regression analysis on the measured VOSO_4_ concentration in the deficient compartment.
(3)$$V_{D}\cdot \frac{d(C_{D}(t))}{dt}=A\cdot \frac{D_{VOSO4}}{L}(C_{E}-C_{D}(t))$$

#### 4.5.4. Oxidative Stability via “Accelerated Stress Test (AST)”

According to a procedure from Oldenburg et al. [36], the oxidative stability was assessed for the following materials: (i) *m*-PBI (2.0 × 4.0 cm, 39 μm thickness, Blue World Technologies, Kvistgård, Denmark), (ii) Fumasep^®^ FAP-450 (2.0 × 4.0 cm, 50 μm thickness, Fumatech BWT Group, Bietigheim-Bissingen, Germany), and (iii) Nafion^®^ NR212 (2.0 × 4.0 cm, 51 μm thickness, Ion Power, New Castle, DE, USA). A solution of 0.2 M cerium sulfate (Ce(SO_4_)_2_ 99%, Acros Organics, Geel, Belgium) in 2 M H_2_SO_4_ at 80 °C was used as agent for the AST. The initial weight of each membrane was measured before starting the test, which lasted for 7 days. Then, the membranes were washed in 2 M H_2_SO_4_ and deionized water to remove any residual of Ce(SO_4_)_2_ and H_2_SO_4_, respectively. Subsequently, they were dried in vacuum at 55 °C for 8 h, after which their weight was measured again. Lastly, the change in weight (Δ*w*) was calculated according to Equation (4). In Equation (4), *w_i_* and *w_f_* are the initial and final weight, respectively.
(4)$$\Delta w=\frac{w_{f}-w_{i}}{w_{i}}\cdot 100\%$$

### 4.6. Cell Operation and Galvanostatic Cycling Tests

All membranes, i.e., (i) Nafion^®^ NR212 (51 μm dry thickness, Ion Power, New Castle, DE, USA), (ii) Fumasep^®^ FAP-450 (50 μm dry thickness, Fumatech BWT Group, Bietigheim-Bissingen, Germany), and (iii) PP-PBI (comprising of a 6 μm *m*-PBI layer and a 30 μm PP layer), were tested in a laboratory electrochemical cell with an active area of 25 cm^2^ using an electrochemical test station (Scribner Model 857 test stand, Scribner Associates, Southern Pines, NC, USA), equipped with in-house designed glass tanks and a multichannel peristaltic pump (Masterflex L/S^®^, GZ-07522-20, Cole-Parmer GmbH, Wertheim, Germany) with plasticizer-free chemical resistant tubing (Versilon™ 2001, GZ-06475-16, Cole-Parmer GmbH, Wertheim, Germany). All data were analyzed using Flowcell^TM^ software (Scribner Associates, Southern Pines, NC, USA) and BView (Scribner Associates, Southern Pines, NC, USA). Prior to assembly in the electrochemical cell, NR212 and FAP-450 membranes were wetted in deionized water, whereas the PP-PBI composite membrane was wetted in IPA due to the hydrophobic nature of the PP support. The PP-PBI membrane was mounted into the electrochemical cell with the *m*-PBI layer facing the negative-side electrolyte and the PP layer facing the positive-side electrolyte.

The electrochemical cell was assembled with (i) two triple-serpentine graphite flow fields (Scribner Associates, Southern Pines, NC, USA), (ii) two gold-plated copper current collectors (Scribner Associates, Southern Pines, NC, USA), (iii) two pre-treated carbon felt electrodes, used as received (25 cm^2^ active area, AAF304ZS, Toyobo, Osaka, Japan), and (iv) two in-house designed polyvinylidene fluoride (PVDF) gasket frames (7.6 × 7.6 cm, 2 mm thickness negative side, 3 mm thickness positive side). The eight bolts of the electrochemical cell were tightened to 4 N·m, with the thickness of the gaskets leading to an average carbon felt electrode compression of 42%.

Next, 1.6 M vanadium in 2 M H_2_SO_4_ and 0.05 M H_3_PO_4_ electrolyte (40 mL for each tank, state of charge of −50%, Oxkem, Reading, UK), constantly purged with argon (5 mL·min^−1^), was used for all cell experiments at an experimentally measured flow rate of 60 mL·min^−1^.

Prior to cycling, the membrane and electrode in the electrochemical cell were conditioned under electrolyte flow (3.5 oxidation state) without any applied current for a period of 4 h. Galvanostatic cycling tests were performed between 0.80 V and 1.65 V as the lower and upper potential limits, respectively. First, the in situ through-plane ohmic area resistance of the cell with the membrane at −50% electrolyte state of charge (SOC) was measured between 100 kHz and 0.1 Hz at zero DC and a perturbation amplitude of 100 mV by electrochemical impedance spectroscopy (EIS). In the same manner, the cell resistance with only TreoPore^®^ PDA-30 was measured. Subsequently, the RFB was charged to 100% SOC at a constant current density of 40 mA·cm^−2^. During the initial charge, polarization curves were recorded at a SOC of 20%, 30%, 50%, 70%, and 90% by consecutively charging and discharging for 20 s at a current density of 20, 40, 60, 80, 100, 120, 150, and 200 mA·cm^−2^. Upon fully charging, five consecutive charge/discharge cycles were performed at 40, 80, 120, 160, and 200 mA·cm^−2^. The first cycle at each current density was not included in the analysis as deviations related to the change in current density could be seen in the data.

Efficiencies and discharge capacity are calculated according to the Equations (5)–(8). In Equations (5)–(7), *Q_ch_* and *Q_dis_* are the charges for the discharge and the charge process, while *V_dis_* and *V_ch_* are the discharge and charge volumes. In Equation (8), *Q_theoretical_* is the theoretical charge, *n* is the number of moles, *F* is the Faraday constant (96,485 C·mol^−1^), and *z* is the charge.
(5)$$\eta_{C}=\frac{Q_{dis}}{Q_{ch}}\cdot 100\%$$
(6)$$\eta_{V}=\frac{V_{dis}}{V_{ch}}\cdot 100\%$$
(7)$$\eta_{E}=(\eta_{C}\cdot \eta_{V})\cdot 100\%$$
(8)$$Q_{theoretical}=I\cdot t=n\cdot (F\cdot z)$$

## 5. Conclusions

The present work demonstrated a composite membrane for VRFBs and a method of preparation. This membrane has an asymmetric configuration with two layers: an *m*-PBI layer (6 µm) and a porous PP layer (30 µm), bonded in an “interlocking” interface, obtained via a gluing procedure by hot-pressing. The *m*-PBI film, known for its stable chemical structure, is able to mitigate V crossover due to its AEM properties in acidic conditions. This asymmetric composite membrane showed the lowest V(IV) diffusivity ((14 ± 1) × 10^−8^ cm^2^·min^−1^) as compared to the commercial Nafion^®^ NR212 and Fumasep^®^ FAP-450, (744 ± 9) × 10^−8^ and (351 ± 1) × 10^−8^ cm^2^·min^−1^, respectively. Furthermore, out of these three membranes, the PP-PBI membrane exhibited the highest energy efficiency (~84%), discharge capacity (~90%), and capacity retention (99%) when galvanostatically charged/discharged over 90 cycles at 120 mA·cm^−2^, making this membrane a promising candidate for the next generation of membranes for RFBs. Nevertheless, more research is required to further enhance the energy efficiency at high current densities by increasing the conductivity or reducing the resistance of the membrane, while maintaining an equal or better capacity retention.

## 6. Patents

J. C. Duburg, A. Schneider, E. Zanzola and L. Gubler, “Method for laminating a polymer electrolyte film onto a porous support layer for energy storage devices”, Paul Scherrer Institute, Switzerland, application no. EP20215599.0, submission date 18 December 2020.

## Figures and Tables

**Figure 1 molecules-26-01679-f001:**
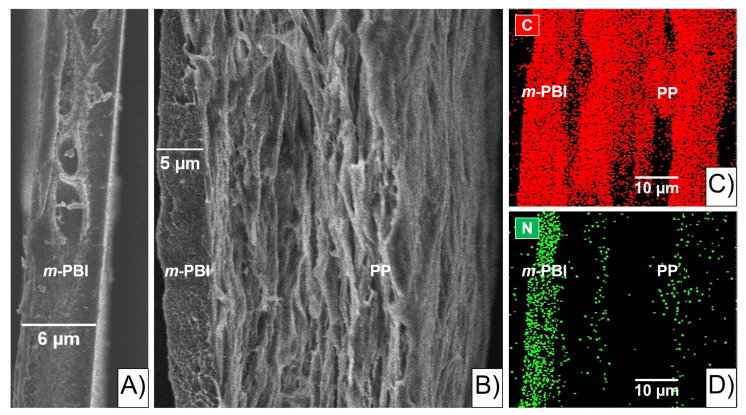
(**A**) Cross-section SEM image (30 μm, 6.0 kV, 2 μA, 9.8 mm working distance (WD), in lens detector) of the *meta*-polybenzimidazole (*m*-PBI) film (6 μm thickness); (**B**) cross-section SEM image (20 μm, 6.0 kV, 2 μA, 9.0 mm WD, in lens detector) of the polypropylene–polybenzimidazole (PP-PBI) composite asymmetric membrane; (**C**,**D**) energy dispersed X-ray (EDX) mapping of the SEM image in (**B**) for carbon and nitrogen (10 μm, 6.0 kV, 2 μA, 9.0 mm WD).

**Figure 2 molecules-26-01679-f002:**
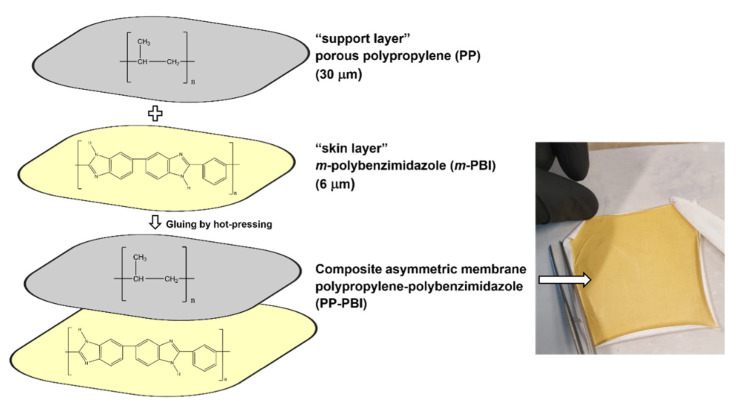
Schematic of the preparation of the asymmetric PP-PBI composite membrane, including a photo of the obtained membrane after hot-pressing.

**Figure 3 molecules-26-01679-f003:**
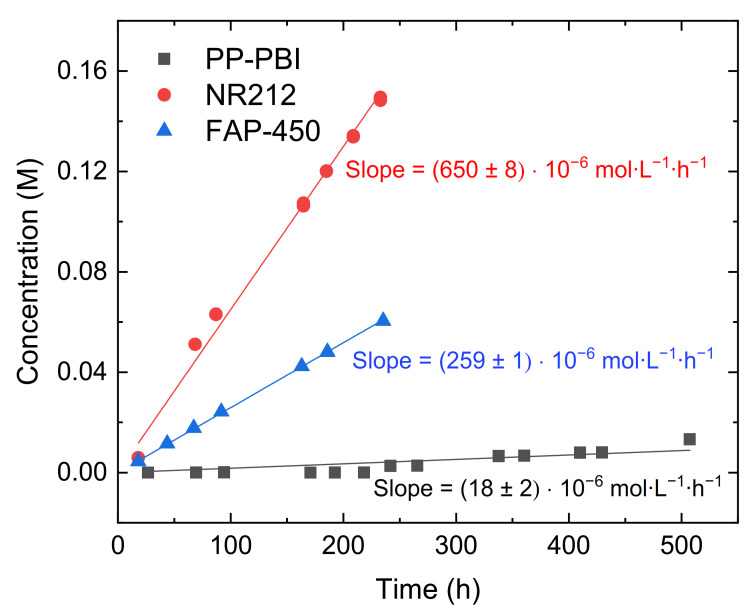
V(IV) diffusion through Nafion^®^ (NR212; red dotted line), Fumasep^®^ (FAP-450; blue dotted line), and PP-PBI (black dotted line).

**Figure 4 molecules-26-01679-f004:**
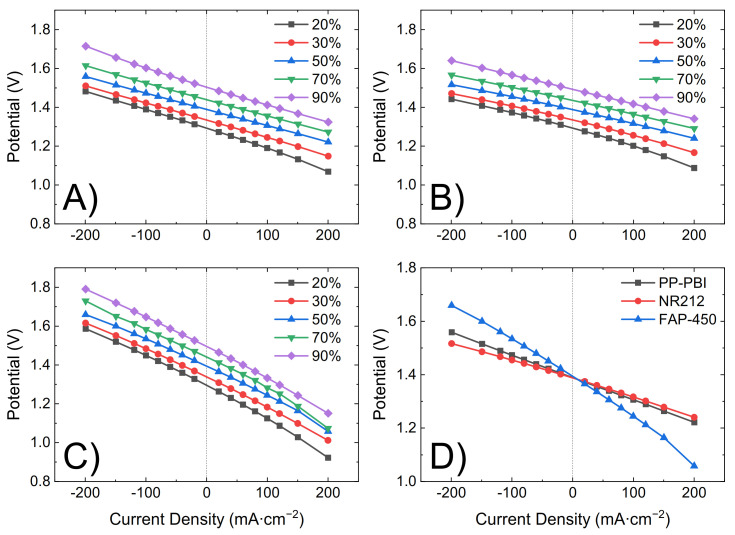
Polarization curves at different electrolyte states of charge (SOCs; 20%, 30%, 50%, 70%, and 90%) for (**A**) PP-PBI, (**B**) NR212, and (**C**) FAP-450. (**D**) Comparison of all three membranes at 50% electrolyte SOC.

**Figure 5 molecules-26-01679-f005:**
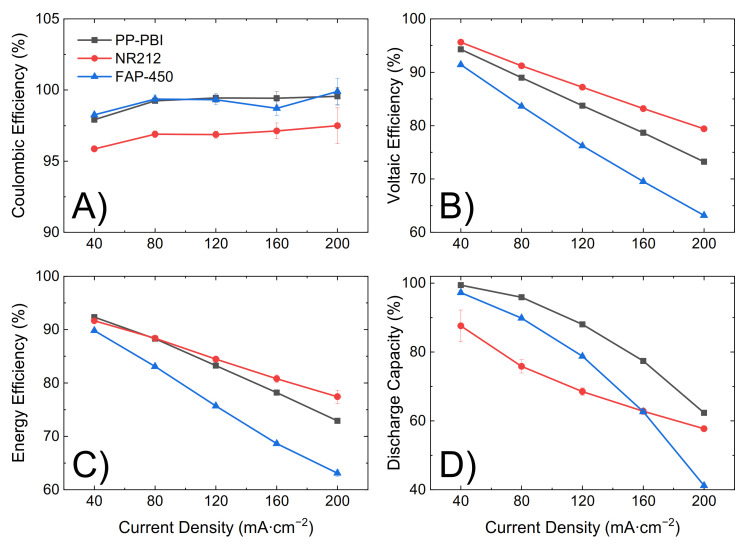
(**A**–**C**) Average efficiency over four cycles for NR212 (red line), FAP-450 (blue line), and PP-PBI (black line) at a current density of 40, 80, 120, 160, and 200 mA·cm^−2^. (**D**) Average discharge capacity over four cycles at a current density of 40, 80, 120, 160, and 200 mA·cm^−2^. The error bars represent the standard deviation over four cycles at each current density.

**Figure 6 molecules-26-01679-f006:**
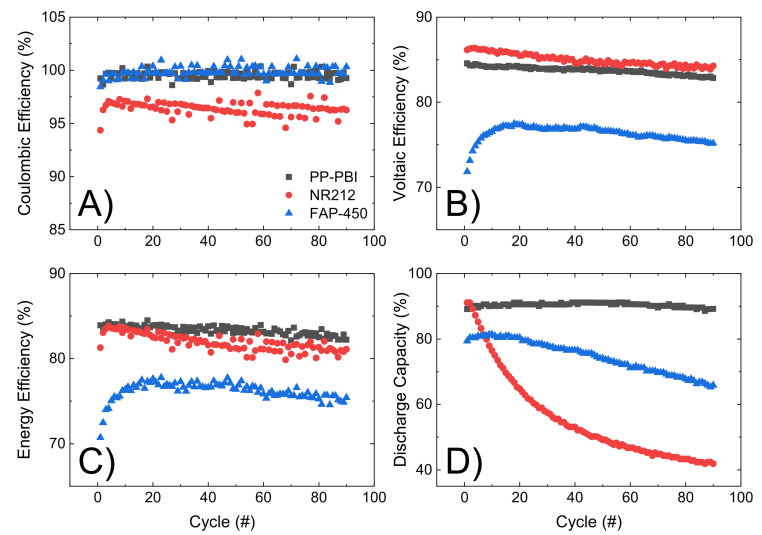
(**A**–**C**) Efficiencies of the galvanostatic cycling at 120 mA·cm^−2^ for NR212 (red dotted line), FAP-450 (blue dotted line), and PP-PBI (black dotted line). (**D**) Discharge capacity for the three membranes. Efficiency values exceeding 100%, likely formed due to the error margin of the measuring equipment (**A**), are not realistic and should be treated as an efficiency of 100%.

**Figure 7 molecules-26-01679-f007:**
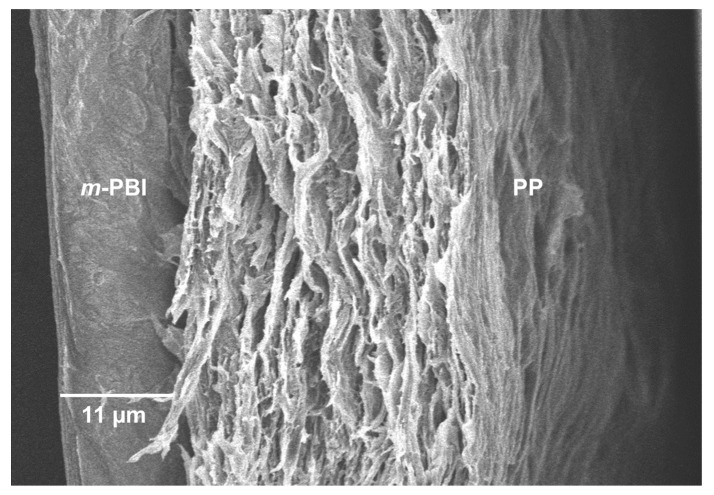
SEM image (30 μm, 5.0 kV, 2 μA, 9.8 mm WD, in lens detector) of cross-section PP-PBI composite membrane post cell cycling at 120 mA·cm^−2^ for 90 cycles.

**Table 1 molecules-26-01679-t001:** V(IV) diffusion through NR212, FAP-450, and PP-PBI.

Name	Slope [V(IV)] vs. *t* (M·L^−1^·h^−1^)	V(IV) Diffusion (cm^2^·min^−1^)
NR212	(650 ± 8) × 10^−6^	(744 ± 9) × 10^−8^
FAP-450	(259 ± 1) × 10^−6^	(351 ± 1) × 10^−8^
PP-PBI	(18 ± 2) × 10^−6^	(14 ± 1) × 10^−8^

**Table 2 molecules-26-01679-t002:** In situ through-plane ohmic area resistance experimentally measured for the cell with TreoPore^®^ PDA-30, NR212, FAP-450, and PP-PBI.

Name	In Situ Through-Plane Ohmic Area Resistance at −50% Electrolyte SOC (Ω·cm^2^)
Cell with only TreoPore^®^ PDA-30 (blank)	0.35
NR212	0.49
FAP-450	1.28
PP-PBI	0.69

## Data Availability

Data available on request. The data will be made available on a publicly accessible repository after the project completion.

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
