# Peer review of "Composite Polybenzimidazole Membrane with High Capacity Retention for Vanadium Redox Flow Batteries"

_molecules, 2021, doi:10.3390/molecules26061679_

Round 1

Reviewer 1 Report

The manuscript by Duburg and co-workers describes the preparation of a polybenzimidazole membrane on a porous support for battery applications, which is a timely and interesting topic. The work has some merits and the results are of interest to the readers of ‘molecules’. There is sufficient data to warrant publication but the context need to be improved and the authors need to improve the description of the work, and justify some of the choices. These and other comments are detailed below, which needs to be addressed prior to further consideration.

1, Where was the porous polypropylene sourced from? Is it a non-woven material? What is the porosity? The authors should give more information on the support.

2, What was the rationale for the gluing method by hot-pressing? Simple blade casting of a PBI dope solution on the PP support would have worked, which is the conventional method for the preparation of this type of membranes. Justification should be provided.

3, What is the molecular weight of the synthesized PBI? Essential characterization for the polymer should be provided given that the authors decided to synthesize their own polymer instead of purchasing commercially available PBI.

4, What are the main pros and cons of the presented membranes? Some critical comments, describing the shortcoming and limitations of the developed materials and methodologies, should be added to the manuscript.

5, Ion-exchange membranes based on PBI and their diverse applications should be acknowledged in the introduction (10.1016/j.memsci.2020.118494, 10.1039/C8TA09160A, 10.1021/acsanm.8b01563, 10.1016/j.memsci.2019.117616).

6, The authors show some error bars on the datasets but it is unclear how these errors were derived. This information should be disclosed in the manuscript. Were independently prepared membranes used for the experiments? How many batches were prepared and analyzed?

7, The active layer is the polybenzimidazole. Why is there a need for the porous polypropylene support?

8, The authors provide three different references for the synthesis of the polybenzimidazole (25-27) but if one protocol was followed than mention that reference, if different components were taken from 3 different references, then specifically mention what was taken from each protocol. (lines 234-235)

9, The conclusion section should have the main results summarized in quantitative statements as well. The authors should clarify how further research and applications will build on the research findings presented here.

Reviewer 2 Report

  1. Authors should review more literatures in the introduction sedciton.
  2. In Fig. 6(a), it is not reasonable coloumbic efficiency at some points is higher than 100%. Authors should provide explaination.
  3. On Page 12, the Nafion NR212 was not numbered. In addition, all membranes should be numbered in the same order throughout this manuscript for consistency.
  4. In section 4.6, Nafion® NR212 and Fumasep® FAP-450 membranes were wetted in deionized water whereas the PP-PBI composite membrane was wetted in IPA. Authors should explain the purpose of different treatments.
  5. In line 388, the treatment method of carbon felt should be described.
  6. Table S1 shows import results and should be shown and discussed in the manuscript.

Round 2

Reviewer 1 Report

The authors have addressed the comments and the manuscript has considerably improved.

Reviewer 2 Report

Authors carefully revised the manuscript. It is acceptable for publication in Molecules.